# A community-powered search of machine learning strategy space to find NMR property prediction models

Lars A. Bratholm[1,2]*, Will Gerrard[1], Brandon Anderson[3,4,5], Shaojie Bai[6,7], Sunghwan Choi[8], Lam Dang[9], Pavel Hanchar[10], Addison Howard[11], Sanghoon Kim[12], Zico Kolter[6,7], Risi Kondor[3,4,13], Mordechai Kornbluth[14], Youhan Lee[15¤], Youngsoo Lee[16], Jonathan P. Mailoa[14], Thanh Tu Nguyen[9], Milos Popovic[17], Goran Rakocevic[17], Walter Reade[11], Wonho Song[18], Luka Stojanovic[17], Erik H. Thiede[3,13], Nebojsa Tijanic[17], Andres Torrubia[19], Devin Willmott[6], Craig P. Butts[1], David R. Glowacki[1,20,21]*

1 School of Chemistry, University of Bristol, Bristol, United Kingdom, 2 School of Mathematics, University of Bristol, Bristol, United Kingdom, 3 Department of Computer Science, The University of Chicago, Chicago, IL, United States of America, 4 Department of Statistics, The University of Chicago, Chicago, IL, United States of America, 5 Atomwise, San Francisco, CA, United States of America, 6 Bosch Center for Artificial Intelligence, Pittsburgh, PA, United States of America, 7 Carnegie Mellon University, Pittsburgh, PA, United States of America, 8 National Institute of Supercomputing and Network, Korea Institute of Science and Technology Information, Yuseong-gu, Daejeon, Republic of Korea, 9 BNP Paribas Cardif, Nanterre Cedex, France, 10 Fyusion, Inc., San Francisco, CA, United States of America, 11 Kaggle, Google Inc., Mountain View, CA, United States of America, 12 Ebay Korea, Gangnam Gu, Seoul, Republic of Korea, 13 Center for Computational Mathematics, Flatiron Institute, New York, NY, United States of America, 14 Bosch Research and Technology Center, Cambridge, MA, United States of America, 15 Department of Chemical and Biomolecular Engineering, Korea Advanced Institute of Science and Technology, Yuseong-gu, Daejeon, Republic of Korea, 16 MINDS AND COMPANY, Gangnam-gu, Seoul, Republic of Korea, 17 Totient Inc, Belgrade, Serbia, 18 KAIST Web Security & Privacy Lab, Yuseong-gu, Daejeon, Republic of Korea, 19 Medbravo.org, Alicante, Spain, 20 Department of Computer Science, University of Bristol, Bristol, United Kingdom, 21 Intangible Realities Laboratory, University of Bristol, Bristol, United Kingdom

¤ Current address: Korea Atomic Energy Research Institute, Yuseong-gu, Daejeon, Republic of Korea
* lars.bratholm@bristol.ac.uk (LAB); drglowacki@gmail.com (DRG)

**Data Availability Statement:** All data is available from the paper, its Supporting Information files, and the following repositories: http://osf.io/kcaht http://github.com/larsbratholm/champs_kaggle.

## Abstract

The rise of machine learning (ML) has created an explosion in the potential strategies for using data to make scientific predictions. For physical scientists wishing to apply ML strategies to a particular domain, it can be difficult to assess in advance what strategy to adopt within a vast space of possibilities. Here we outline the results of an online community-powered effort to swarm search the space of ML strategies and develop algorithms for predicting atomic-pairwise nuclear magnetic resonance (NMR) properties in molecules. Using an open-source dataset, we worked with Kaggle to design and host a 3-month competition which received 47,800 ML model predictions from 2,700 teams in 84 countries. Within 3 weeks, the Kaggle community produced models with comparable accuracy to our best previously published 'in-house' efforts. A meta-ensemble model constructed as a linear combination of the top predictions has a prediction accuracy which exceeds that of any individual model, 7-19x better than our previous state-of-the-art. The results highlight the potential of transformer architectures for predicting quantum mechanical (QM) molecular properties.

**Funding:** WG is partially supported by the EPSRC National Productivity Investment Fund (NPIF) for Doctoral Studentship funding. LAB thanks the Alan Turing Institute under the EPSRC grant EP/N510129/1. DRG is supported by the Leverhulme Trust (Philip Leverhulme Prize) and Royal Society (URF/R/180033). LAB and DRG acknowledge support of this work through the "CHAMPS" EPSRC programme grant EP/P021123/1. SC was supported by National Research Foundation of Korea (2018R1D1A1B07049981, 2019M3E5D4065968) funded by the Ministry of Science and ICT. Authors SB, LD, PH, AH, SK, ZK, MK, YL, JPM, TTN, MP, GR, WR, LS, NT, and DW are affiliated with commercial companies. The funders had no role in study design, data collection and analysis, decision to publish, or preparation of the manuscript. The specific roles of these authors are articulated in the 'author contributions' section.

**Competing interests:** Authors SB, LD, PH, AH, SK, ZK, MK, YL, JPM, TTN, MP, GR, WR, LS, NT, and DW are affiliated with commercial companies. The funders had no role in study design, data collection and analysis, decision to publish, or preparation of the manuscript. This does not alter our adherence to PLOS ONE policies on sharing data and materials. There are no patents, products in development or marketed products associated with this research to declare.

## 1. Introduction

The rise of machine learning (ML) in the physical sciences has created a number of notable successes [1–7], and the number of published outputs is increasing substantially [8]. This explosion is perhaps not entirely surprising, given that ML 'search space' is effectively infinite. For example, the performance of a particular ML algorithm strategy depends sensitively on at least four components: (a) the dataset used for training (and the corresponding methodology used for dataset curation); (b) the feature selection used to construct ML inputs; (c) the choice of ML algorithm; and (d) the values of the optimal constituent hyperparameters. For components (b) and (c), the space of possibilities is continually expanding; for components (a) and (d), the space of possibilities is potentially infinite. Given the sensitivity of ML approaches to each of the items outlined above, ML's explosion within the scientific literature has led to warnings of an emerging computational reproducibility crisis, a risk exacerbated by the fact that many peer-reviewed ML publications do not include the data and algorithms required to reproduce their results [9].

The difficulty of searching an enormous ML space is compounded by the fact that the training of even simple neural networks has been shown to be an NP-complete problem [10]. Deciphering whether any global optima lurk within an effectively infinite ML search space has been the topic of a great deal of research; however, there seems to be a consensus emerging that it is practically impossible to demonstrate that any particular ML strategy is in fact optimal or bias-free, even for very simple systems [11]. Broadly speaking, the parameter spaces in which a particular ML strategy can be constructed are non-convex, and characterized by multiple local minima and saddle points in which optimization algorithms can get trapped [12]. Nevertheless, ML algorithms can produce useful results. In a nod to the 1950 Japanese period drama "Rashomon" (where various characters provide subjective, alternative, self-serving, yet compelling versions of the same incident), ML's tendency to produce many accurate-but-different models has been referred to as the "Rashomon effect" in machine learning [13]. In such a vast space, any individual agent has a chance of stumbling upon a reasonable ML model. Given the difficulty of rationalizing the uniqueness of any particular ML model or approach, individual models are increasingly being used as constituents within ensemble models, whose combined accuracy outperforms that of any individual model [14].

Over the last several years, a number of studies have demonstrated the utility of 'crowd-sourced' approaches for solving scientific problems which involve searching hyperdimensional spaces [15–19]. Inspired by recent attempts within both particle physics [20, 21] and materials science [22] using community power to develop ML algorithms, we worked with Kaggle (an online platform for ML competitions), to design a competition encouraging participants to develop ML models able to accurately predict QM nuclear magnetic resonance (NMR) properties from 3D molecular structure information [23]. The fact that some of our authorship team had worked in this area over several years [24] meant that we had quantitative and qualitative benchmarks to analyse competition progress in relation to what conventional academic research approaches had achieved. The so-called 'Champs Kaggle Competition' (CKC) ran from 29-May-19 through 28-Aug-19. The 5 models which achieved the highest accuracy were awarded respective prize money of $12.5k, $7.5k, $5k, $3k and $2k. Over ~13 weeks, the CKC received 47,800 model predictions from 2,700 teams in 84 countries (Fig 1A), representing the most exhaustive search to date of ML strategies aimed at predicting QM NMR properties from 3D molecular structure information. The number of participants who engaged with the CKC was amongst the highest for any physical science challenge which Kaggle has hosted to date. Fig 1B and 1C show a steady increase in the number of participants who joined the CKC versus time. CKC participants reported being drawn to the competition because it: (a) facilitated

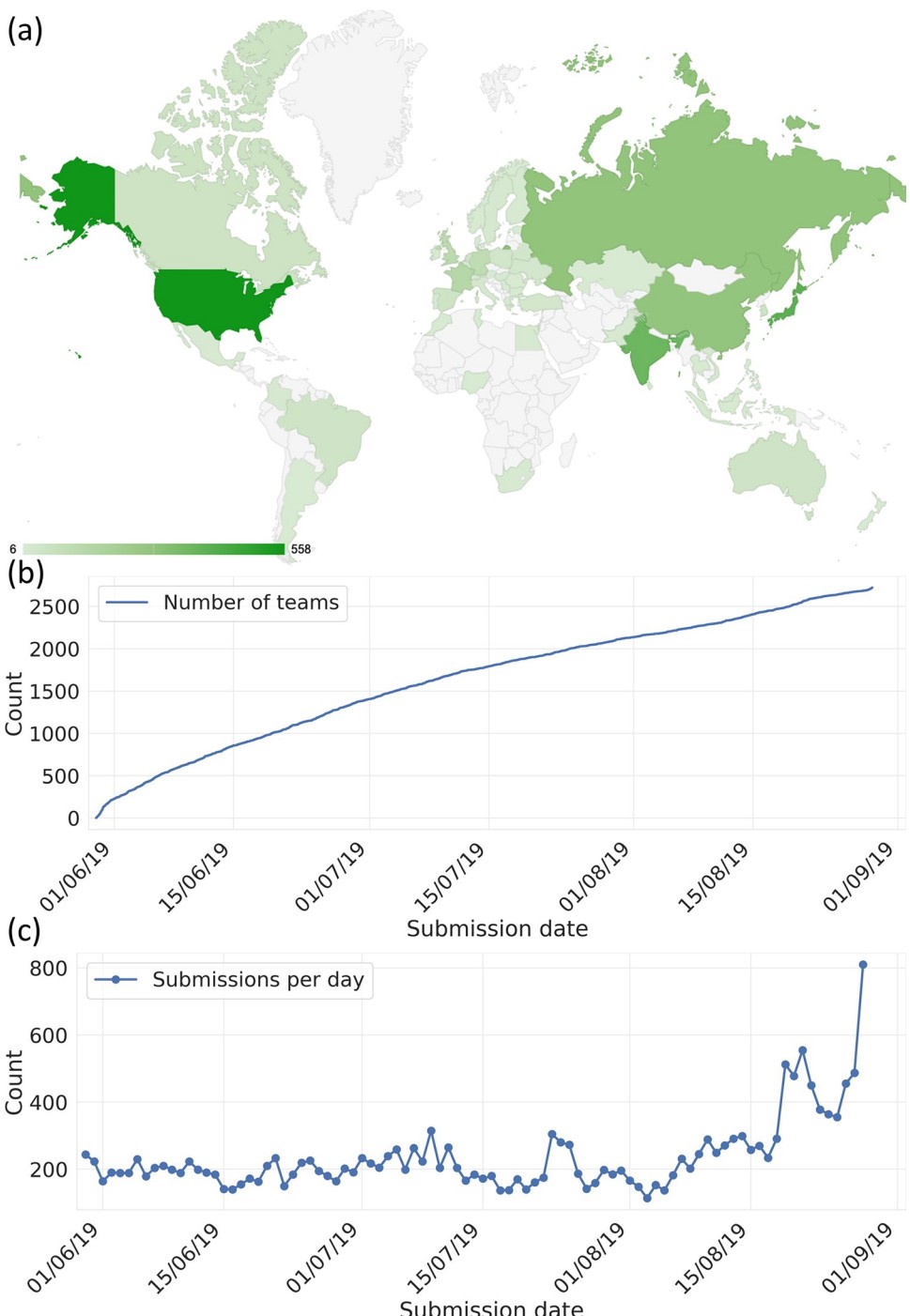

**Fig 1.** (a) map showing the number teams participating from different countries over the duration of the CKC (countries with less than 5 participants are shown light gray); (b) the number of CKC participants vs. time; (c) number of submissions per day.

progress on an important research problem; (b) involved a rich, noise-less dataset whose structure was easy to understand; and (c) had a dataset which was manageable using standard data processing tools, workflows, and hardware.

## 2. Competition design

### 2.1. Domain

NMR is the dominant spectroscopic technique for determining 2D and 3D molecular structure in solution. Amongst the most important data obtained in an NMR spectrum are the chemical shifts (which describe the position/frequency of a signal in the spectrum) and the scalar couplings (which determine the splitting/shape of the signal in the spectrum). ML methods to predict NMR properties are established in academic and commercial workflows for determining 2D molecular structure from experimental NMR datasets [25–28]. Despite this success, these 2D approaches often fail when the NMR properties are affected by 3D structure, for example atoms are separated by several bonds yet remain close in 3D space (ring current effects, hydrogen bonding etc.). This is an inherently difficult problem as the 3D molecular structure is simply not well described by 2D representations and there are not enough high-quality experimental data available to accurately infer most 3D relationships from a 2D structural representation alone.

The most accurate computed predictions of NMR properties use QM methods like density functional theory (DFT) to get a one-to-one mapping between a 3D structure and the contribution it has to the experimentally observed NMR property. Accurate QM methods for NMR property predictions are powerful but expensive. Recent work has thus focussed on developing ML algorithms which can efficiently reproduce the results of costly QM methods, achieving results in seconds rather than hours or days [24, 29]. ML approaches have the added appeal that they can be trained using large datasets of DFT-computed NMR parameters, which are not limited to experimental structural observations. With a large enough training database, we have shown in previous work that an ML strategy can approach the accuracy of DFT calculations of atom-centered NMR parameters such as chemical shift for 3D structure analysis, but with several orders of magnitude reduction in time [24].

Beyond NMR, the last decade has seen considerable effort focused on machine learning QM molecular properties [30–36]. Broadly speaking, this work has tended to focus on predicting atomic properties such as partial charges, or molecular properties such as energies and dipoles. Relatively little work has been carried out designing ML models which are able to predict pairwise atomic properties such as scalar coupling constants. Our earlier work to develop pairwise property prediction algorithms were effectively independent-atom treatments, in which atomic feature vectors describing the local environment of each atom were concatenated [24]. However, this approach loses information about the relative position/orientation of each atom's respective environment, which is important for multiple-bond couplings. The CKC represents an attempt to kickstart research into ML methods able to make accurate prediction of pairwise properties.

### 2.2. Dataset & scoring

Scalar couplings are critically dependent on the 3D structure of the molecule for which they are being measured; however at the time we carried out this work, we were unaware of accurate experimental databases linking pair-wise mutiple-bond NMR scalar couplings to well-defined 3D molecular structures. Therefore, we decided to run the CKC utilizing molecular structures included in the QM9 dataset, a publicly available benchmark for developing ML models of 3D structure-property relationships [37]. QM9 includes ~134k molecules comprised of carbon, fluorine, nitrogen, oxygen and hydrogen. The molecules included within QM9 have no more than 9 heavy atoms (non-hydrogen), with a maximum of 29 total atoms. To obtain a corresponding set of scalar couplings, we extended the QM9 computational methodology,

using the B3LYP functional [38] and the 6-31g(2df,p) basis set [39–42] to compute NMR parameters on the optimized QM9 structures. The computed QM9 scalar coupling constants are available under Creative Commons CC-NC-BY 4.0, enabling others to build on this work.

To remove the possibility of CKC participants overfitting their models to the entire set of computed QM9 scalar couplings, 65% of molecules in the dataset were randomly partitioned into a training set and the other 35% to a testing set. The test set was further split, with 29% of the data in a 'public' test set, and 71% of the data in a 'private' test set (competitors were unaware of the specifics of the private/public split). Both the training and test sets included the molecular geometries and indices of the coupling atoms. Unlike the test set, the training set included a range of other data, including the calculated scalar coupling values, their break-down into Fermi contact (FC), spin-dipole (SD), paramagnetic spin-orbit (PSO) and diamagnetic spin-orbit (DSO) components, and a range of auxiliary information obtained from the QM computations (e.g., potential energy, dipole moment vectors, magnetic shielding tensors and Mulliken charges). As the CKC progressed, participating teams continually iterated and improved their models. A regularly updated and publicly visible leaderboard enabled each team to see where their model ranked in predicting the public test set data compared to the model predictions made by all of the other teams.

The leaderboard scores were determined using a function which accounted for the 8 different types of coupling constants included in the training and testing datasets: $^1J_{HC}$, $^1J_{HN}$, $^2J_{HH}$, $^2J_{HC}$, $^2J_{HN}$, $^3J_{HH}$, $^3J_{HC}$ and $^3J_{HN}$ (where the superscript indicates the number of covalent bonds separating the atom pairs indicated by the subscript). Since the number of couplings of each type differed (e.g., the molecular composition of the QM9 test set included 811,999 $^3J_{HC}$ couplings compared to 24,195 $^1J_{HN}$ couplings) and spanned different value ranges, the scoring function used the average of the logarithm of the mean absolute error for each type of coupling constant:

$$\text{score} = \frac{1}{T}\sum_{t}^{T} \log\left(\frac{1}{n_t}\sum_{i}^{n_t} |y_i - \hat{y}_i|\right) \tag{1}$$

where $t$ is an index that runs over the $T = 8$ different scalar coupling types, $i$ is an index that spans $1..n_t$, the number of observations of type $t$, $y_i$ is the scalar coupling constant for observation $i$, and $\hat{y}_i$ is the predicted scalar coupling constant for observation $i$. This scoring function ensures, for example, that a 10% improvement in one type of coupling will improve the score by the same amount as a 10% improvement in another type of coupling, so that no coupling class dominates.

## 2.3. Recruitment & consent

The competition was run using the online competition platform Kaggle. Recruitment was done via Kaggle marketing. All participants consented to the competition rules (see S14 in S1 Appendix) prior to submitting solutions. We provided information to the participants about the overall purpose of the competition (e.g., to develop new quantum mechanical property predication algorithms, and to aid design of medicines. See S15 in S1 Appendix for full description used), the timeline, submission format and objective scoring metric, and further-more by answering questions on the discussion forum [43]. Participants were not required to provide us with any additional information. Upon competition completion, the email addresses of the competition winners were passed to the main authors/organizers in order to invite their collaboration on this paper (all members of the winning teams are co-authors). While the scoring metric used to determine winners were objective, since the data set used were synthetic, there was a risk that a team could infer the computational methodology and perform well by cheating. However, to be eligible for a cash prize, the winners had to share

their code that could reproduce their predictions. This helped ensure that the competition was resistant to any form of cheating.

## 3. Results

### 3.1. Leaderboard time evolution

Over the course of the CKC, Fig 2A shows the evolution of the best score whose source code was publicly available (public notebooks), and its relationship to the top score versus time. Fig 2B shows that the time trace of the top score is well fit by a bi-exponential curve with two distinct phases. **Phase 1** lasted for the first week, during which time the accuracy increased by ~12x (~2.5 improvement in score), with a time constant of ~1.29 ± 0.18 days. **Phase 2** lasted for the next 12 weeks, during which time the accuracy improved more gradually by a factor of ~4x (~1.5 improvement in score), with a time constant of ~50.0 ± 16.6 days. To determine which models were awarded prize money, the final set of model rankings were assessed using Eq (1) to evaluate how well each of the models predicted the scalar coupling values in the private test set (preventing competitors inferring the target property from the leaderboard scores rather than from the training set). Due to the large amount of noise-less data, the positioning in the top 37 submissions was the same on the public and private leaderboard at the end of the CKC. Several teams commented that the stability between the public & private leaderboards made for an enjoyable competition.

The top-scoring method achieved a geometric mean error (which is the exponential of the score in Eq (1)) of 0.039 Hz which was 6-16x more accurate than what could be achieved using our own recently developed methodology (see S5 and S13 in S1 Appendix for details) [24]. In addition to the final score, Kaggle also rewards participants who make the best contributions to: (1) publicly available code, and (2) the discussion forums. As a result of these incentives, a number of participants opted to voluntarily publish their source code (public notebooks). In many cases, the public notebooks were then utilized and adapted by other CKC participants. As shown in Fig 2A, the best score achieved using these public notebooks follows a time trace which is similar to the leading score, but less accurate by ~1.5. A number of participants made instructional web posts, scripts, and videos outlining specific approaches which they had taken during the CKC. For example, video presentations by Andrey Lukyanenko [44] and the NVIDIA team [45] discuss the approaches which they utilized to develop the 8[th] and 33[rd] place solutions, respectively. The CKC summary features insightful write-ups by several top teams in which they describe their various model approaches [46].

### 3.2. Meta-ensemble model

To assess the extent to which the prize-winning submissions differed from one another (and other highly ranked submissions), we used the top 400 submissions to construct a meta ensemble (ME) model as a linear combination of the top scoring models:

$$y_{i,ME} = \sum_{j=k}^{400} w_j y_{i,j} \tag{2}$$

Given that many of the top models (and all of the prize winners) were ensemble models, we have adopted the term "meta-ensemble" (ME) to emphasize the fact that Eq (2) is an ensemble of ensemble models. In Eq (2), the ME prediction $y_{i,ME}$ of the $i$'th scalar coupling constant is a linear combination of the predictions $y_{i,j}$ of the $j$'th ranked model. The index $k$ specifies the lowest ranked model to be included within the optimized ME model. When $k = 1$, Eq (2) runs over the entire list of the top 400 models. When $k > 1$, Eq (2) neglects top-scoring models. Setting $k = 6$ for example, the Eq (2) ME model excludes all of the prize-winning models (ranks

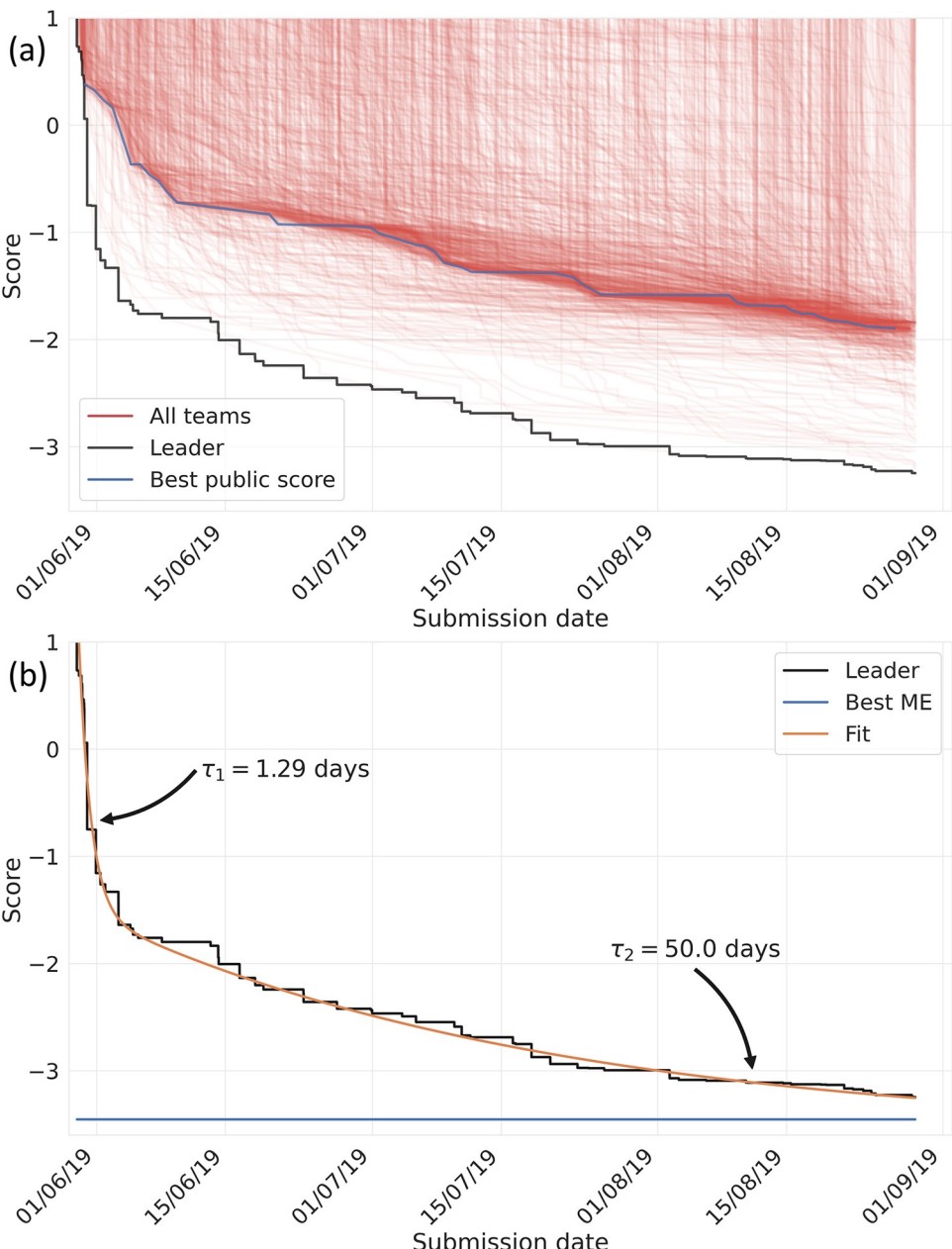

**Fig 2.** (a) score evolution vs. time. Black line shows the best performing method vs. time. Blue line shows the best performing public notebook. Red lines shows the best submission by each team; (b) best fit of the time dependent leader (black) score to an biexponential curve of the form $A \cdot exp(-t/\tau_1) + B \cdot exp(-t/\tau_2) + C$ ($A = 2.11$; $B = 2.97$; $\tau_1 = 50.0$ days; $\tau_2 = 1.29$ days; $C = -3.59$). Blue indicates the best ME model score.

#1 –#5). For ME models constructed using Eq (2), the weights $w_j$ were determined by minimizing $y_{i,ME}$ using half of the test set, under the constraint that the weights were positive and summed to unity. While a range of different ME models can be constructed (e.g., different ensembles for each type of coupling, median averaging etc.), this simple mean is easy to interpret.

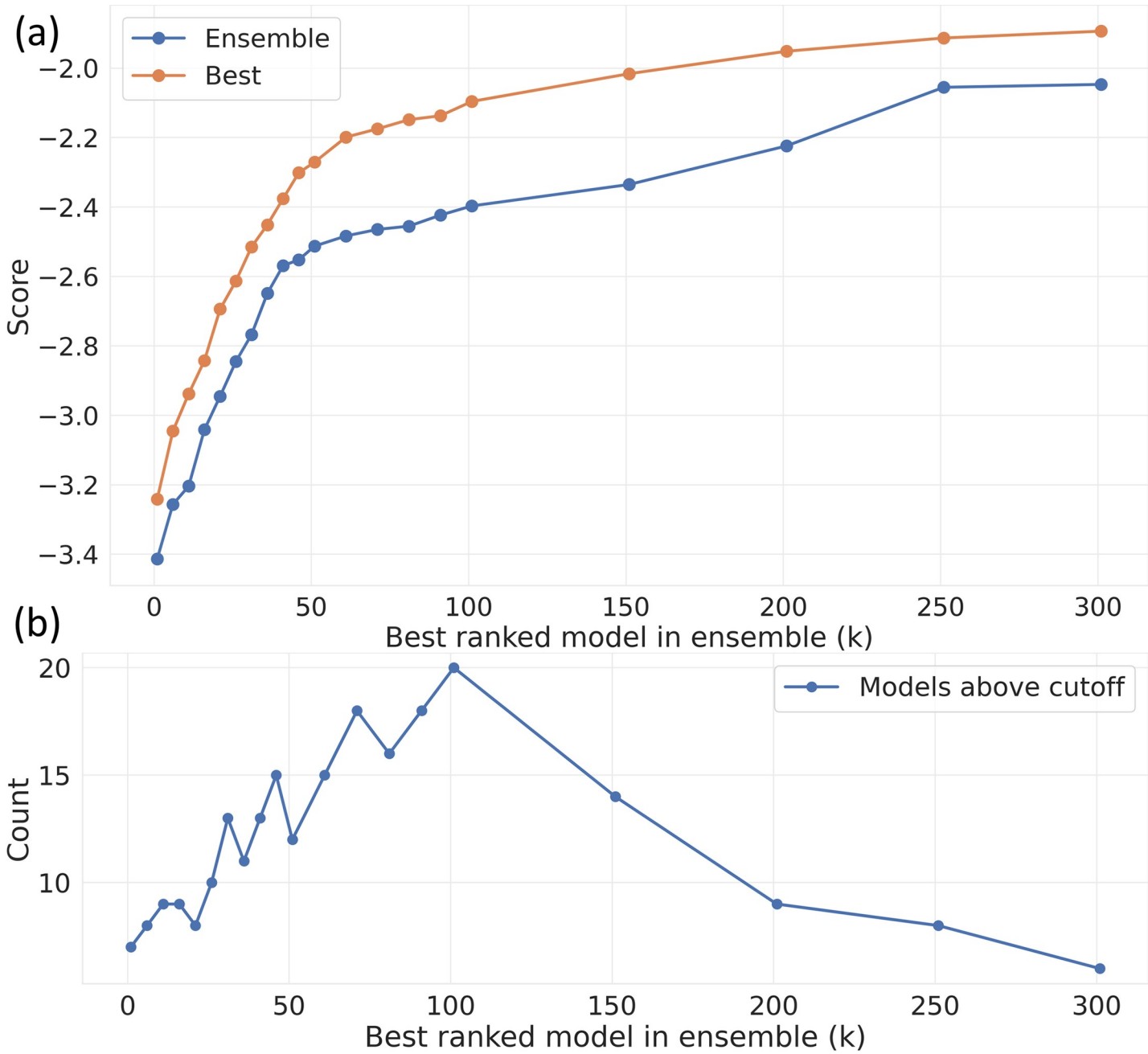

**Fig 3.** (a) comparison of the top individual score (orange) to the ME model score (blue) as function of the k value in Eq (2); (b) the number of contributors to the ME model at a particular k value that had an optimized weight greater than 0.01.

Different classes of machine learning algorithms (or even the same algorithm with different hyperparameters) may be able to learn different regions of the data better than others. Thus, by combining the highest scoring model predictions that have the least correlation for a meta-ensemble, the strengths of various models may be accumulated, a result confirmed by the ME analysis shown in Fig 3A as a function of $k$. As expected (for $k = 1..300$) the optimized ME model achieves an accuracy which always surpasses that of the best individual model. In the regime where the top scorers are incrementally being

**Table 1. Summary descriptions for the six models in the final ME.**

| | 1st | 2nd | 3rd | 4th | 5th | 12th |
|---|---|---|---|---|---|---|
| Weight $w_i$ | 0.204 | 0.270 | 0.111 | 0.203 | 0.046 | 0.149 |
| Number of submissions | 73 | 167 | 151 | 37 | 53 | 4 |
| Country | USA | Spain<br>Belarus | S. Korea | Serbia | France | USA |
| Team size | 5 | 2 | 5 | 4 | 3 | 2 |
| Any Chemistry expertise? | Y | N | Y | Y | N | Y |
| Use of scalar coupling components? | N | N | Y | N | N | N |
| Translational invariance? | Y | Y | N | Y | Y | Y |
| Rotational invariance? | Y | N | N | Y | Y | Y |
| Previous Kaggle experience? | N | Y | Y | N | Y | N |
| Included additional input features? | Y | N | N | Y | Y | N |
| Number of model parameters | ~105M | ~60M | ~70M | ~60M | ~66M | ~250K |

"Use of Scalar coupling components" refers to whether a team decomposed the scalar couplings into four separate components in their model.

eliminated from Eq (2) ($k = 1..50$), Fig 3A shows that the ME model has a score that is ~0.2 lower than the "best" model. For example, the $k = 7$ ME model (which neglects the top 6 models) still outperforms the winning solution, and the $k = 11$ ME model outperforms the winning solution when the per-type ensemble mentioned above is used. Fig 3B shows how many contributors to the ME model at a particular k value had an optimized weight greater than 0.01. Broadly speaking, the Fig 3A results can be lumped into three regimes. In the first regime ($k \sim 1..40$) the best performing methods dominate the ME and there is little to be gained by including within the ME methods that are very different if they perform worse. In the second regime ($k \sim 41..200$), Fig 3A shows that the gap between the top score and the ME model widens to ~0.4. Here there are many similarly performing yet different methods, so there is much to be gained by combining their different approaches into a ME. In the third regime ($k \sim 201..300$) the gain from a ME decreases, presumably because many of the models are similar variants of the public notebooks. The relative benefit of constructing a ME model (versus using a top-scoring model) thus appears to be more significant outside of the band of top-scoring and low-scoring models.

For the $k = 1$ ME model, which was 7-19x more accurate than our previously published model [24], we analysed in further detail its constituents. The results in Table 1 show the $k = 1$ ME constituents with weights $w_i > 0.02$, along with the relative rankings $j$ of the constituent ME models. Table 1 shows that there is no particular model which is dominant: there are five models with a weighting greater than 0.11, and three with a weighting greater than 0.20. Of the six models in Table 1, one (#12) falls outside the top 5. Its 0.149 contribution is larger than prize winning models #3 and #5. Fig 4A shows the submission history of the Table 1 models, and their relationship to the overall public leader board.

### 3.3. Correlation analysis

To further understand the relationship between the winning submissions within the $k = 1$ ME model, we carried out a correlation analysis on the top 50 team submissions. The submissions were then ordered using a hierarchical clustering analysis (see S5 and S13 in S1 Appendix). The results in Fig 4B show that the #1 –#5 teams are part of the same sub-cluster *i.e.* all relatively similar to each other. Fig 4C specifically highlights the low correlation between models #1 –#5 compared to model #12, which shows that this team's approach exists within a region

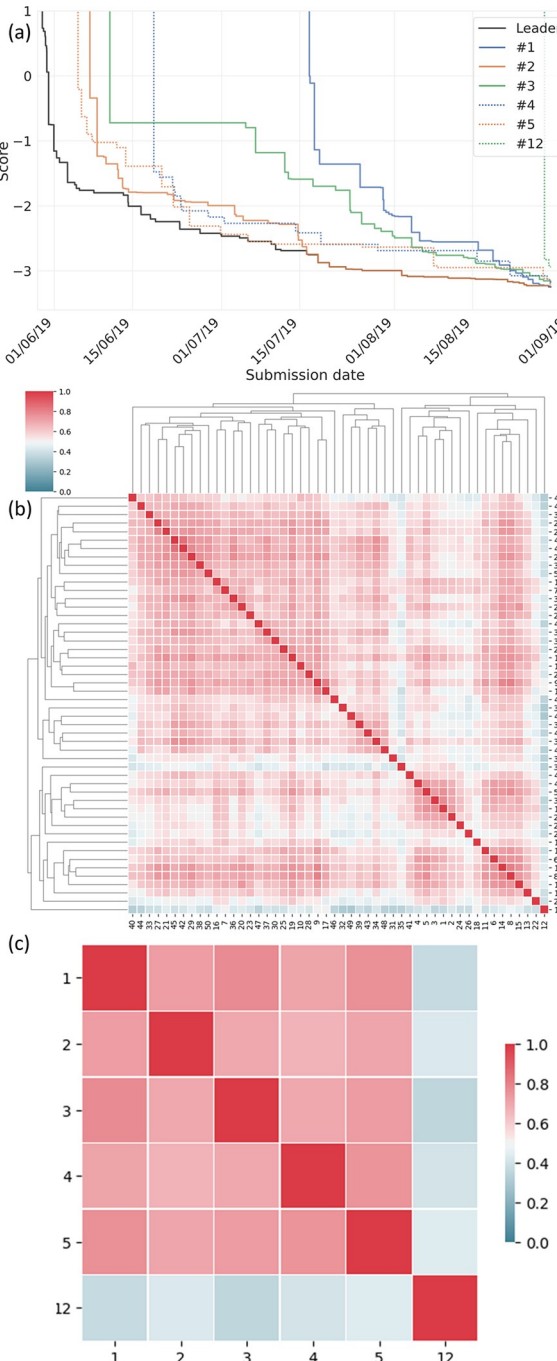

**Fig 4.** (a) score evolution vs. time for **Table 1** teams. Black line shows the best performing method at a current time. Colored lines show the best submission by each team; (b) correlation amongst the top 50 submissions. Red indicates high correlation, and blue low. Bottom and right side shows the ranking of the submission, while top and left features a dendrogram depicting the hierarchical clustering; (c) correlation between Table 1 teams.

of ML strategy space that appears relatively distinct from the prize-winning models, and also from the top 50 solutions.

Compared to the others in Table 1, team #12 was a relative latecomer to the CKC as shown in Fig 4A. In addition, the number of parameters in their model is ~100x smaller than the

others. The low correlation of team #12 compared to the other teams in Table 1 appears to have arisen because they utilized the 'Cormorant' rotationally covariant neural network strategy [47]. Originally developed for learning molecular potential energy surfaces (PESs), Cormorant takes advantage of rotational symmetries in order to enforce physical relationships in the resultant neural network, by using spherical tensors to encode local geometric information around each atom's environment, which transform in a predictable way under rotation. The use of spherical tensors allows for a network architecture that is covariant to rotations, so that if a rotation is applied to a layer, all activations at the next layer will automatically inherit that rotation. As such, a rotation to a Cormorant input will propagate through the network to ensure that the output transforms as well. This captures local geometric information while still maintaining the desired transformation properties under rotations. Team #12's sophisticated input processing strategy contrasts with the approaches taken by other teams, which tended to utilize far simpler encoding strategies, either by restricting the input features to have translational and rotational invariance (e.g., using internal distances), adding translational and rotational noise to make the inputs robust to rotation and translation, or allowing the model learn invariance on its own. Team #12's approach is grounded in domain specific physics knowledge, and characteristic of the emphases which physical scientists tend to apply in ML contexts.

## 4. Discussion & conclusions

Community-powered approaches offer a powerful tool for searching ML strategy space and providing accurate predictions for physical science problems like the prediction of 2-body QM NMR properties. Within 3 weeks, the best score on the Kaggle public leader board achieved an accuracy which surpassed our own previously published approaches [24], suggesting that an open source community-powered 'swarm search' of ML strategy space may in some cases be significantly faster and more cost-efficient than conventional academic research strategies where a single agent (e.g., a PhD student or post-doctoral researcher) spends several years hunting for solutions in an infinite search space. ME model construction combined with correlation analysis highlights the strength of the CKC 'swarm search' approach, in line with the "Rashomon effect".

Whereas our earlier approaches to predicting NMR structure coupling constants [24] had relied on kernel-ridge regression approaches [48] where the internal distances and angles in the molecules were systematically encoded to a feature vector for the coupling atom pairs using predetermined basis functions, the community which emerged around the CKC pioneered a new application of transformer neural nets [49] to QM molecular property prediction. While such networks have found extensive use for sequence modelling and transduction problems such as language modelling and machine translation, they represent a relatively new approach to predicting QM properties like NMR shifts or scalar couplings, and it will be interesting to explore their further application to other QM properties and more general 2-body property prediction problems, which are relevant in several domains across the physical sciences. The rich portfolio of open source blog posts, data, insight, source code, and discussions arising from the CKC offers an excellent foundation for subsequent research and follow-up studies, through community initiatives or more conventional academic research approaches.

Teams #2 and #5 had no domain specific expertise, and yet outperformed participants with domain expertise, including our own previous attempts [24]. This contrasts with previously published Kaggle competitions in particle physics [20, 21] and materials science [22], where the winners tended to be domain experts. Table 1 shows that teams with prior domain expertise (e.g., #1 and #4) used their insight to calculate additional input features beyond those

which we provided, and which they then used as model input. For example, team #1 used Mulliken charges and atomic valency, while team #4 used electronegativity, first ionization energy, electron affinity, mulliken charge, and bond types. Despite this added complexity, team #1 only narrowly managed (i.e., within the CKC's final hours) to improve on the approach of team #2, which used a simple cartesian input representation with no additional data.

All of the prize winning teams utilized deep neural networks where the encoder learned the pair-feature vectors from the coordinates, atom types, distances, etc. A separate feed forward neural network (decoder) was then used to make scalar coupling predictions per coupling type or sub coupling type. The relatively simple input descriptions used by many of the top teams transferred to the neural network the challenge of learning an effective input representation. Such approaches contrast with those favored by physical scientists, which utilize more complex descriptors constructed so as to include domain specific insight (e.g., rotational symmetries for team #12). Taking advantage of the variance in approaches, the various model predictions can be combined into a ME model whose combined accuracy surpasses that of any individual model, 7-19x more accurate than what our previous methods were able to achieve. The benefit of a ME model seems to be most significant in the regime where there are many independent individual models with similar performance.

Fig 2A shows that the average benefit which new models contributed to the overall improvement in prediction accuracy decreased versus time, with a rapid improvement over the first week, followed by a much more gradual improvement over the next 13 weeks. Fig 1C shows that the number of model predictions was approximately constant versus time with an increase over the final 20 days. These observations indicate an overall decrease in the relative cost/benefit ratio as a function of time. This cost/benefit decrease is qualitatively compatible with conclusions drawn from previous meta-analyses of scientific progress [50], which suggest that search strategies for scientific discovery tend to become less efficient with time. In our case, these results suggest that a shorter competition may have furnished similar insights. The results also highlight potential shortcomings in the elaborate scheme of awards and prizes which scientific disciplines utilize to incentivize progress and recognize 'top-performers'–e.g., the fact that solution #12 played a more important role in the optimized ME model compared to some of the prize winning models offers an important reminder that scientific progress is a community effort that depends on a range of important contributions, which can often go unrecognized in conventional indicators of prestige.

The results of this study demonstrate how community science initiatives in conjunction with open data can enable rapid scientific progress in ML domains, reaffirming the community benefits that can arise when scientific workers make their data and algorithms open. Web-based platforms enable distributed community efforts to build engagement with scientific concepts at a time where scientific approaches face mounting challenges across media and political landscapes. Given the constraints on conventional scientific collaboration which have arisen as a result of social distancing, distributed scientific community efforts like these may become more prevalent in the near term. For example, there has been a steady increase in the number of scientific stack exchanges, which (like Kaggle) incentivize scientific communities to share knowledge and expertise. Digital platforms which benefit from the ubiquity of cloud computing and which enable distributed communities to engage with one another to undertake collective problem solving are likely to play an important role in our emerging scientific future. Such approaches may be particularly useful for problems like ML, where the strategy spaces are effectively infinite. Moving forward, it will be interesting to explore the extent to which search efficiencies might be enhanced by combining the intelligence of human agents with machine agents.

## Supporting information

**S1 Appendix. Dataset generation and contents, additional results analysis, model architectures, directions to code and competition description.**
(PDF)

## Acknowledgments

Part of this work was carried out using the computational facilities of the Advanced Computing Research Centre, University of Bristol. We thank Prof. Jan H. Jensen for access to additional computing resources and Dr. Christopher Sutton for helpful advice on the design of the CKC. We thank the contributions to the wider competition of all the hundreds of Kaggle participants (listed at www.kaggle.com/c/champs-scalar-coupling/leaderboard) and especially Guillaume Huard for support with this work.

## Author Contributions

**Conceptualization:** Lars A. Bratholm, Craig P. Butts, David R. Glowacki.

**Data curation:** Lars A. Bratholm.

**Formal analysis:** Lars A. Bratholm, Will Gerrard, Brandon Anderson, Shaojie Bai, Sunghwan Choi, Lam Dang, Pavel Hanchar, Sanghoon Kim, Zico Kolter, Mordechai Kornbluth, Youhan Lee, Youngsoo Lee, Jonathan P. Mailoa, Thanh Tu Nguyen, Milos Popovic, Goran Rakocevic, Walter Reade, Wonho Song, Luka Stojanovic, Erik H. Thiede, Nebojsa Tijanic, Andres Torrubia, Devin Willmott.

**Funding acquisition:** Addison Howard, Craig P. Butts, David R. Glowacki.

**Investigation:** Lars A. Bratholm, Will Gerrard, Brandon Anderson, Shaojie Bai, Sunghwan Choi, Lam Dang, Pavel Hanchar, Sanghoon Kim, Zico Kolter, Mordechai Kornbluth, Youhan Lee, Youngsoo Lee, Jonathan P. Mailoa, Thanh Tu Nguyen, Milos Popovic, Goran Rakocevic, Wonho Song, Luka Stojanovic, Erik H. Thiede, Nebojsa Tijanic, Andres Torrubia, Devin Willmott.

**Methodology:** Lars A. Bratholm, Brandon Anderson, Shaojie Bai, Sunghwan Choi, Lam Dang, Pavel Hanchar, Sanghoon Kim, Zico Kolter, Risi Kondor, Mordechai Kornbluth, Youhan Lee, Youngsoo Lee, Jonathan P. Mailoa, Thanh Tu Nguyen, Milos Popovic, Goran Rakocevic, Wonho Song, Luka Stojanovic, Erik H. Thiede, Nebojsa Tijanic, Andres Torrubia, Devin Willmott, David R. Glowacki.

**Project administration:** Lars A. Bratholm, Addison Howard, David R. Glowacki.

**Software:** Lars A. Bratholm, Brandon Anderson, Shaojie Bai, Sunghwan Choi, Lam Dang, Pavel Hanchar, Sanghoon Kim, Zico Kolter, Risi Kondor, Mordechai Kornbluth, Youhan Lee, Youngsoo Lee, Jonathan P. Mailoa, Thanh Tu Nguyen, Milos Popovic, Goran Rakocevic, Walter Reade, Wonho Song, Luka Stojanovic, Erik H. Thiede, Nebojsa Tijanic, Andres Torrubia, Devin Willmott.

**Supervision:** Craig P. Butts, David R. Glowacki.

**Validation:** Lars A. Bratholm.

**Visualization:** Lars A. Bratholm, Brandon Anderson, Shaojie Bai, Sunghwan Choi, Lam Dang, Pavel Hanchar, Sanghoon Kim, Zico Kolter, Mordechai Kornbluth, Youhan Lee, Youngsoo Lee, Jonathan P. Mailoa, Thanh Tu Nguyen, Milos Popovic, Goran Rakocevic, Walter

Reade, Wonho Song, Luka Stojanovic, Erik H. Thiede, Nebojsa Tijanic, Andres Torrubia, Devin Willmott.

**Writing – original draft:** Lars A. Bratholm, David R. Glowacki.

**Writing – review & editing:** Lars A. Bratholm, Will Gerrard, Brandon Anderson, Shaojie Bai, Sunghwan Choi, Lam Dang, Pavel Hanchar, Addison Howard, Sanghoon Kim, Zico Kolter, Mordechai Kornbluth, Youhan Lee, Youngsoo Lee, Jonathan P. Mailoa, Thanh Tu Nguyen, Milos Popovic, Goran Rakocevic, Walter Reade, Wonho Song, Luka Stojanovic, Erik H. Thiede, Nebojsa Tijanic, Andres Torrubia, Devin Willmott, Craig P. Butts, David R. Glowacki.

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
