## [Decision Letter · Decision Letter 0]

18 Mar 2021

PONE-D-21-02243

A community-powered search of machine learning strategy space to find NMR property prediction models

PLOS ONE

Dear Dr. Bratholm,

Thank you for submitting your manuscript to PLOS ONE. After careful consideration, we feel that it has merit but does not fully meet PLOS ONE’s publication criteria as it currently stands. Therefore, we invite you to submit a revised version of the manuscript that addresses the points raised during the review process.

The reviewers have raised some minor issues; please respond to each of them by revision or rebuttal.

We look forward to receiving your revised manuscript.

Kind regards,

Dennis Salahub

Academic Editor

PLOS ONE

Journal Requirements:

2. Please include information about the competition, participant recruitment and participant consent in the Methods section. Please include details regarding participant consent to collect personal data, including email addresses, names, or phone numbers. In the Methods section, please ensure that you have specified how consent was obtained and how the study met relevant personal data and privacy laws. If data were collected anonymously, please include this information.

"I have read the journal's policy and the authors of this manuscript have the following competing interests: This competition was made possible through financial support from Kaggle, where AH and WR are employees."

We note that one or more of the authors are employed by a commercial company: BNP Paribas Cardif, Fyusion Inc.,  Kaggle, Google Inc., & Ebay Korea, MINDS AND COMPANY, Totient Inc.

3.1. Please provide an amended Funding Statement declaring this commercial affiliation, as well as a statement regarding the Role of Funders in your study. If the funding organization did not play a role in the study design, data collection and analysis, decision to publish, or preparation of the manuscript and only provided financial support in the form of authors' salaries and/or research materials, please review your statements relating to the author contributions, and ensure you have specifically and accurately indicated the role(s) that these authors had in your study. You can update author roles in the Author Contributions section of the online submission form.

3.2. Please also provide an updated Competing Interests Statement declaring this commercial affiliation along with any other relevant declarations relating to employment, consultancy, patents, products in development, or marketed products, etc.  

4. We note that Figure 1 in your submission contain map images which may be copyrighted. All PLOS content is published under the Creative Commons Attribution License (CC BY 4.0), which means that the manuscript, images, and Supporting Information files will be freely available online, and any third party is permitted to access, download, copy, distribute, and use these materials in any way, even commercially, with proper attribution. For these reasons, we cannot publish previously copyrighted maps or satellite images created using proprietary data, such as Google software (Google Maps, Street View, and Earth). For more information, see our copyright guidelines: http://journals.plos.org/plosone/s/licenses-and-copyright.

4.1.    You may seek permission from the original copyright holder of Figure 1 to publish the content specifically under the CC BY 4.0 license. 

4.2.    If you are unable to obtain permission from the original copyright holder to publish these figures under the CC BY 4.0 license or if the copyright holder’s requirements are incompatible with the CC BY 4.0 license, please either i) remove the figure or ii) supply a replacement figure that complies with the CC BY 4.0 license. Please check copyright information on all replacement figures and update the figure caption with source information. If applicable, please specify in the figure caption text when a figure is similar but not identical to the original image and is therefore for illustrative purposes only.

5. One of the noted authors is a group or consortium [Kaggle participants]. In addition to naming the author group, please list the individual authors and affiliations within this group in the acknowledgments section of your manuscript. Please also indicate clearly a lead author for this group along with a contact email address.

Reviewers' comments:

Reviewer's Responses to Questions

**Comments to the Author**

1. Is the manuscript technically sound, and do the data support the conclusions?

Reviewer #1: Yes

Reviewer #2: Yes

2. Has the statistical analysis been performed appropriately and rigorously? 

Reviewer #1: N/A

Reviewer #2: Yes

3. Have the authors made all data underlying the findings in their manuscript fully available?

Reviewer #1: Yes

Reviewer #2: Yes

4. Is the manuscript presented in an intelligible fashion and written in standard English?

Reviewer #1: Yes

Reviewer #2: Yes

5. Review Comments to the Author

Reviewer #1: This is an interesting report on the use of a crowd competition using Kaggle to produce and compare machine learning models for prediction of NMR coupling constants. It is a useful example and generally well explained and on the whole well analysed in the paper.

While I understand why the comparison is with computed structures and QM calculations it is a pity that experimental work can not be included and a proper assessment of the importance of experimental uncertainty included in the study.

Specific comments

The scoring function given in equation (1) is fine and justified but then I was confused to see a discussion of the geometric mean (exponential of the score??) a bit further down in the paper.

The way in which remodels converged over the time fo the competition is interesting but I don't think worth fitting the exponentials and making quite such a fuss about as the paper is more about the models not human behaviour (though there is certainly a paper in that as well). So I suggest that either lessor more eon this is discussed.

The use of the meta ensemble is good but the whole discussion of the range of ranked members of the ensemble and the weights etc come across as rather confused. I am sure this an be simplified and made much clearer. The point that different models while displaying different scores are sensitive to different cases is interesting and very well worth exploring.

The tone of the discussion comes over and rather dismissive of the physics inspired models, perhaps because they are better. I also suggest the use of physics is not necessarily the best way to describe some thing that is just as much chemistry, even if it is in common use.

Reviewer #2: The manuscript describes an attempt to create a collaborative effort using a prize/award type of incentive and demonstrates its potential to overcome the local minimum problem in non-convex search spaces. I find the manuscript well written and refreshing as it demonstrate the virtues of collaboration, openness and that diversity matters.

The manuscript focuses on the prediction of NMR scalar coupling constants that are computationally expensive to obtain by ab-initio techniques. The results are convincing as the combination of some of the top performers allowed for an increase in accuracy by an order of magnitude. Thus, I warmly recommend this manuscript for publication.

As a final comment I admit that as a chemist involved in computational model I would have appreciated a more chemical discussion about the type of errors encountered in relation to the molecular species or conformations.

A minor detail, but in the document submitted the correlation matrix b) is useless as numbering cannot be read.

6. PLOS authors have the option to publish the peer review history of their article (what does this mean?). If published, this will include your full peer review and any attached files.

Reviewer #1: No

Reviewer #2: No

---

## [Author Response · Author response to Decision Letter 0]

8 May 2021

See 'Response to Reviewers.pdf' for reviewer comments and requested formatting changes.

---

## [Decision Letter · Decision Letter 1]

9 Jun 2021

A community-powered search of machine learning strategy space to find NMR property prediction models

PONE-D-21-02243R1

Dear Dr. Bratholm,

We’re pleased to inform you that your manuscript has been judged scientifically suitable for publication and will be formally accepted for publication once it meets all outstanding technical requirements.

Kind regards,

Dennis Salahub

Academic Editor

PLOS ONE

Additional Editor Comments (optional):

Reviewers' comments:

Reviewer's Responses to Questions

**Comments to the Author**

1. If the authors have adequately addressed your comments raised in a previous round of review and you feel that this manuscript is now acceptable for publication, you may indicate that here to bypass the “Comments to the Author” section, enter your conflict of interest statement in the “Confidential to Editor” section, and submit your "Accept" recommendation.

Reviewer #1: All comments have been addressed

2. Is the manuscript technically sound, and do the data support the conclusions?

Reviewer #1: Yes

3. Has the statistical analysis been performed appropriately and rigorously? 

Reviewer #1: Yes

4. Have the authors made all data underlying the findings in their manuscript fully available?

Reviewer #1: Yes

5. Is the manuscript presented in an intelligible fashion and written in standard English?

Reviewer #1: Yes

6. Review Comments to the Author

Reviewer #1: I consider that the authors have made sufficient changes to answer the referees’ comments and I therefore recommend publication of the revised manuscript.

7. PLOS authors have the option to publish the peer review history of their article (what does this mean?). If published, this will include your full peer review and any attached files.

Reviewer #1: No

---

## [Editor Report · Acceptance letter]

12 Jul 2021

PONE-D-21-02243R1 

A community-powered search of machine learning strategy space to find NMR property prediction models 

Dear Dr. Bratholm:

I'm pleased to inform you that your manuscript has been deemed suitable for publication in PLOS ONE. Congratulations! Your manuscript is now with our production department. 

Kind regards, 

on behalf of

Dr. Dennis Salahub 

Academic Editor

PLOS ONE